# MiniGPT-5: Interleaved Vision-and-Language Generation via Generative Vokens

## Abstract

Large Language Models (LLMs) have garnered significant attention for their advancements in natural language processing, demonstrating unparalleled prowess in text comprehension and generation. Yet, the simultaneous generation of images with coherent textual narratives remains an evolving frontier. In response, we introduce an innovative interleaved vision-and-language generation technique anchored by the concept of "generative vokens," acting as the bridge for harmonized image-text outputs. Our approach is characterized by a distinctive two-staged training strategy focusing on description-free multimodal generation, where the training requires no comprehensive descriptions of images. To bolster model integrity, classifier-free guidance is incorporated, enhancing the effectiveness of vokens on image generation. Our model, MiniGPT-5, exhibits substantial improvement over the baseline Divter model on the MMDialog dataset and consistently delivers superior or comparable multimodal outputs in human evaluations on the VIST dataset, highlighting its efficacy across diverse benchmarks.

## 1 Introduction

In the recent development of larger-scale vision-and-language models, multi-modal feature integration is not just a passing trend but a critical advancement shaping a wide array of applications, from multimodal dialogue agents to cutting-edge content creation tools. With the surge in research and development, vision-and-language models such as (Wu et al., 2023; Zhu et al., 2023) are on the brink of an era where they're expected to comprehend and generate both text and image content seamlessly. This multi-faceted ability is crucial, as it fosters enhanced interactions across various domains like virtual reality, media, and e-commerce. Essentially, the task is to enable models to coherently synthesize, recognize, and respond using both visual and textual modalities, harmonizing the information flow and creating cohesive narratives. However, as we tread the path towards blending textual and visual modalities and achieving the envisioned interleaved vision-and-language generation, as illustrated in 1, we recognize that it's driven by the pressing need for more integrated and fluid multimodal interactions in today's large language models. Yet, this journey is riddled with obstacles.

The challenges posed by multimodal generation are manifold. For starters, there exists a dichotomy between state-of-the-art Large Language Models (LLMs) (OpenAI, 2023; Chiang et al., 2023) capabilities. While they excel in understanding text and processing text-image pairs, they falter in the nuanced art of generating images. This limitation becomes even more glaring when the data itself lacks comprehensive descriptions. Moving away from conventional tasks that benefited from exhaustive image descriptions, the emerging interleaved vision-and-language tasks (Wang et al., 2020; Sharma et al., 2018) lean heavily on topic-centric data, often skimping on thorough image descriptors (Huang et al., 2016). Then there's the intrinsic challenge associated with end-to-end models like BLIP-2 (Li et al., 2023b) and MiniGPT-4 (Zhu et al., 2023). Even after being trained on massive datasets, aligning generated text with corresponding images is an intricate puzzle. Lastly, as we push the boundaries with LLMs, the towering memory requirements beckon us to devise more efficient strategies, especially in downstream tasks.

Addressing these nuanced challenges, our proposed methodology charts a promising trajectory. By amalgamating the Stable Diffusion mechanism with LLMs through special visual tokens (Tan & Bansal, 2020) – "generative vokens", we herald a new pattern for proficient multimodal genera-

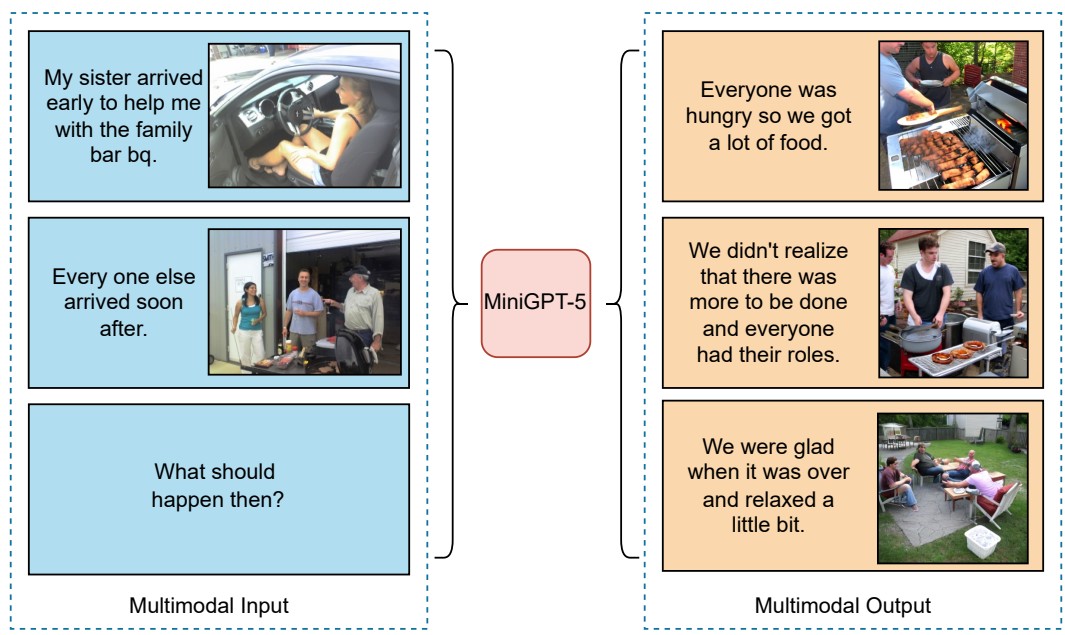

Figure 1: MiniGPT-5 is a unified model for interleaved vision-and-language comprehension and generation. Besides the original multimodal comprehension and text generation abilities, MiniGPT-5 can provide appropriate, coherent multimodal outputs.

tion. Our pioneering two-stage training methodology underlines the importance of a description-free foundational phase, prepping the model to thrive even in data-scarce scenarios. Our generic stages, free from domain-specific annotations, make our solution distinct from the existing body of work. To ensure that the generated text and images are in harmony, our dual-loss strategy comes into play, further enhanced by our innovative generative voken approach and classifier-free guidance. To round off, our parameter-optimized fine-tuning approach grapples with memory constraints, optimizing training efficiency.

Building on these techniques, our work signifies a transformative approach. As shown in Figure 2, using ViT (Vision Transformer) and Qformer (Li et al., 2023b) along with the large language models, we transmute multimodal inputs into generative vokens, seamlessly pairing with the high-resolution Stable Diffusion 2 model (Rombach et al., 2021) for context-aware image generation. Incorporating images as auxiliary input with instruction tuning approaches and pioneering both the text and image generation loss, we amplify the synergy between text and visuals. Our proposed MiniGPT-5, set against models like CLIP's constraints (Rombach et al., 2021), masterfully fuses diffusion models with MiniGPT-4, delivering unmatched multi-modal results without domain-specific annotation dependence. Crucially, our strategy can capitalize on advancements in multimodal vision-language foundational models, holding promising prospects for enhanced multimodal generative prowess.

Our contribution are three folds:

- We propose to use multimodal encoders representing a novel and generic technique that has proved more efficient than LLM and also inversion to generative vokens, and combine it with Stable DIffusion to generate interleaved vision-and-language outputs. [multimodal language model that can do multimodal generation]

- We highlight a new two-staged training strategy for the description-free multimodal generation. The unimodal alignment stage harvests the high-quality text-aligned visual features from large text-image pairs. The multimodal learning stage includes a novel training task prompted context generation, ensuring the visuals and text prompt can well coordinate for generation. The inclusion of classifier-free guidance during the training phase further refines generation quality.

- Compared with other multimodal generation models, we achieved state-of-the-art performance on the CC3M dataset. We also established unprecedented benchmarks on prominent datasets, including VIST and MMDialog.

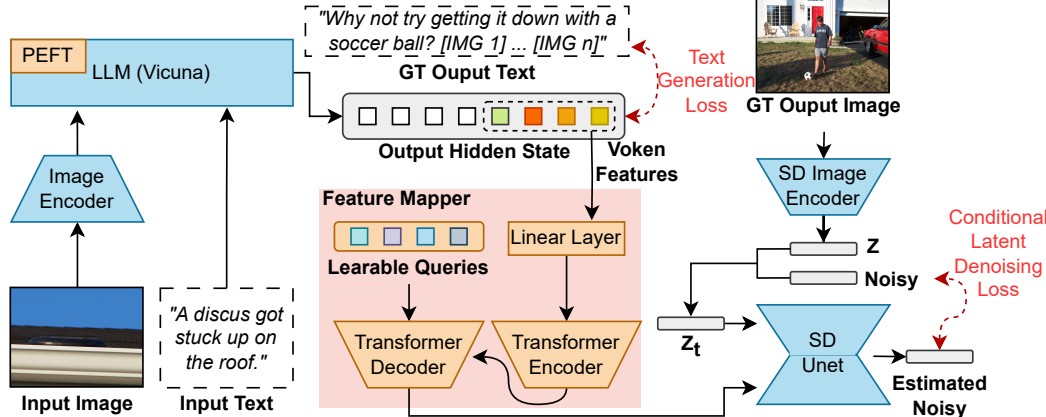

Figure 2: The overview structure of MiniGPT-5 pipeline. We leverage the pretrained multimodal large language model (MiniGPT-4) and text-to-image generation model (Stable Duffision 2) to create a unified multimodal generation pipeline. The input image encoder includes a ViT, Qformer, and linear layer, pretrained by MiniGPT-4. The orange blocks include learnable parameters, while the blue blocks are fixed during training. More details can be find in Section 2.

## 2 METHOD

In order to endow large language models with multimodal generation capabilities, we introduce a structured framework that integrates pretrained multimodal large language models and text-to-image generation models. To address the discrepancies across model domains, we introduce special visual tokens—termed generative vokens—that are able to direct training on raw images. Moreover, we advance a two-stage training methodology, coupled with a classifier-free guidance strategy, to further enhance the quality of generation. Subsequent sections will provide a detailed exploration of these elements.

### 2.1 MULTIMODAL INPUT PROCESS

Recent advancements in multimodal large language models, such as MiniGPT-4, have primarily concentrated on multimodal comprehension, enabling the processing of images as sequential input. To expand their capabilities to multimodal generation, we introduce generative vokens designed for outputting visual features. Additionally, we employ cutting-edge, parameter-efficient fine-tuning techniques within the Large Language Model (LLM) framework for multimodal output learning. A more detailed introduction to these developments will be provided in the following paragraphs.

**Multimodal Encoding:** Each text token is embedded into a vector $e_{\text{text}} \in \mathbf{R}^d$, while the pretrained visual encoder transforms each input image into the feature $e_{\text{img}} \in \mathbf{R}^{32 \times d}$. These embeddings are concatenated to create the input prompt features.

**Adding Vokens in LLM:** Since the original LLM's $V$ vocabulary only includes the textual tokens, we need to construct a bridge between the LLM and the generative model. Therefore, we introduce a set of special tokens $V_{\text{img}} = \{[\text{IMG1}], [\text{IMG2}], \ldots, [\text{IMGn}]\}$ (default $n = 8$) as generative vokens into the LLM's vocabulary $V$. The LLM's output hidden state for these vokens is harnessed for subsequent image generation, and the positions of these vokens can represent the insertion of the interleaved images. With all pretrained weights $\theta_{\text{pretrained}}$ in MiniGPT-4 fixed, the trainable parameters include extra input embedding $\theta_{\text{voken\_input}}$ and output embedding $\theta_{\text{voken\_output}}$.

**Parameter-Efficient Fine-Tuning (PEFT):** Parameter-efficient fine-tuning (PEFT) (Houlsby et al., 2019) is critical in training large language models (LLMs) like ChatGPT (OpenAI, 2023). Despite this, its application in multimodal settings remains largely unexplored. We use PEFT over the MiniGPT-4 (Zhu et al., 2023) encoder to train a model to understand instructions or prompts better, enhancing its performance in novel and even zero-shot tasks. More specifically, we tried prefix tuning (Li & Liang, 2021) and LoRA over the entire language encoder — Vicuna (Chiang et al.,

2023) used in MiniGPT-4. Combined with the instruction tuning, it notably amplifies multimodal generation performance across various datasets, such as VIST and MMDialog.

## 2.2 MUTIMODAL OUPUT GENERATION

To accurately align the generative tokens with the generative model, we formulate a compact mapping module for dimension matching and incorporate several supervisory losses, including text space loss and latent diffusion model loss. The text space loss assists the model in learning the correct positioning of tokens, while the latent diffusion loss directly aligns the tokens with the appropriate visual features. Since the generative vokens' features are directly guided by images, our method does not need comprehensive descriptions of images, leading to description-free learning.

**Text Space Generation:** We first jointly generate both text and vokens in the text space by following the casual language modeling. During the training, we append the vokens to the positions of ground truth images and train the model to estimate vokens within text generation. Specifically, the generated tokens are represented as $T = \{t_1, t_2, \ldots, t_m\}$, where $t_i \in V \cup V_{\text{img}}$, and the causal language modeling loss is defined as:

$$L_{\text{text}} := -\sum_{i=1}^{m} \log p(t_i | e_{\text{text}}, e_{\text{img}}, t_1, \ldots, t_{i-1}; \theta_{\text{pretrained}}, \theta_{\text{voken\_input}}, \theta_{\text{voken\_output}}), \text{ where } t_i \in V \cup V_{\text{img}}$$

$$(1)$$

**Mapping Voken Features for Image Generation:** Next, we align the output hidden state $h_{\text{voken}}$ with the text conditional feature space of the text-to-image generation model. To map the voken feature $h_{\text{voken}}$ to a feasible image generation conditional feature $e_{\text{text\_encoder}} \in \mathbf{R}^{L \times \hat{d}}$ (where $L$ is the maximum input length of text-to-image generation text encoder, and $\hat{d}$ is the dimension of encoder output feature in text-to-image generation model), we construct a feature mapper module, including a two-layer MLP model $\theta_{\text{MLP}}$, a four-layer encoder-decoder transformer model $\theta_{\text{enc-dec}}$, and a learnable decoder feature sequence $q$. The mapping feature $\hat{h}_{\text{voken}}$ is then given by:

$$\hat{h}_{\text{voken}} := \theta_{\text{enc-dec}}(\theta_{\text{MLP}}(h_{\text{voken}}), q) \in \mathbf{R}^{L \times \hat{d}}$$

$$(2)$$

**Image Generation with Latent Diffusion Model (LDM):** To generate appropriate images, the mapping feature $\hat{h}_{\text{voken}}$ is used as a conditional input in the denoising process. Intuitively, $\hat{h}_{\text{voken}}$ should represent the corresponding text features that guide the diffusion model to generate the ground truth image. We employ the loss of the latent diffusion model (LDM) for guidance. During the training, the ground truth image is first converted to latent feature $z_0$ through the pretrained VAE. Then, we obtain the noisy latent feature $z_t$ by adding noise $\epsilon$ to $z_0$. A pretrained Unet model $\epsilon_\theta$ is used to calculate the conditional LDM loss as:

$$L_{LDM} := \mathbb{E}_{\epsilon \sim \mathcal{N}(0,1),t} \left[ \left\| \epsilon - \epsilon_\theta \left( z_t, t, \hat{h}_{\text{voken}} \right) \right\|_2^2 \right]$$

$$(3)$$

This comprehensive approach ensures a coherent understanding and generation of both textual and visual elements, leveraging the capabilities of pretrained models, specialized tokens, and innovative training techniques.

## 2.3 TRAINING STRATEGY

Given the non-negligible domain shift between text and image domains, we observe that direct training on a limited interleaved text-and-image dataset can result in misalignment and diminished image quality. Consequently, we adopt two distinct training strategies to mitigate this issue. The first strategy encompasses the incorporation of the classifier-free guidance technique, which amplifies the effectiveness of the generative tokens throughout the diffusion process. The second strategy unfolds in two stages: an initial pre-training stage focusing on coarse feature alignment, followed by a fine-tuning stage dedicated to intricate feature learning.

**Classifier-free Guidance (CFG):** To enhance the coherence between the generated text and images, we first leverage the idea of Classifier-free Guidance for multimodal generation. Classifier-free guidance is introduced in the text-to-image diffusion process. This method observes that the

generation model $P_\theta$ can achieve improved conditional results by training on both conditional and unconditional generation with conditioning dropout. In our context, since the generation model is fixed, our objective is to accentuate the trainable condition $h_{\text{voken}}$. During training, we replace $h_{\text{voken}}$ with zero features $h_0 \in \mathbf{0}^{n \times d}$ with a 10% probability, obtaining the unconditional feature $\hat{h}_0 = \theta_{\text{enc-dec}}(\theta_{\text{MLP}}(h_0), q)$. During inference, $\hat{h}_0$ serves as negative prompting, and the refined denoising process is expressed as:

$$
\begin{aligned}
\log \widehat{\mathrm{P}_\theta} \left( \epsilon_t \mid z_{t+1}, \hat{h}_{\text{voken}}, \hat{h}_0 \right) = {}& \log \mathrm{P}_\theta \left( \epsilon_t \mid z_{t+1}, \hat{h}_0 \right) + \\
& \gamma \left( \log \mathrm{P}_\theta \left( \epsilon_t \mid z_{t+1}, \hat{h}_{\text{voken}} \right) - \log \mathrm{P}_\theta \left( \epsilon_t \mid z_{t+1}, \hat{h}_0 \right) \right)
\end{aligned}
\tag{4}
$$

**Two-stage Training Strategy:** Recognizing the non-trivial domain shift between pure-text generation and text-image generation, we propose a two-stage training strategy: Unimodal Alignment Stage (**UAS**) and Multimodal Learning Stage (**MLS**). Initially, we align the voken feature with image generation features in single text-image pair datasets, such as CC3M, where each data sample only contains one text and one image and the text is usually the caption of the image. During this stage, we utilize captions as LLM input, enabling LLM to generate vokens. Since these datasets include the image descriptive information, we also introduce an auxiliary loss to aid voken alignment, minimizing the distance between the generative feature $\hat{h}_{\text{voken}}$ and the caption feature from the text encoder $\tau_\theta$ in the text-to-image generation model:

$$
L_{\text{cap}} := \text{MSE}(\hat{h}_{\text{voken}}, \tau_\theta(c))
\tag{5}
$$

The unimodal alignment stage loss is expressed as $L_{\text{UAS}} = \lambda_1 * L_{\text{text}} + \lambda_2 * L_{\text{LDM}} + \lambda_3 * L_{\text{cap}}$, with selected values $\lambda_1 = 0.01, \lambda_2 = 1, \lambda_3 = 0.1$ to rescale the loss into a similar numerical range.

After the unimodal alignment stage, the model is capable of generating images for single text descriptions but struggles with interleaved vision-and-language generation, which includes multiple text-image pairs and requires complicated reasoning for both text and image generation. To address this, in the multimodal learning stage, we further fine-tune our model with PEFT parameters by interleaved vision-and-language datasets, such as VIST, where the data sample has several steps with text-image and texts are sequentially relevant. During this stage, we construct three types of tasks from the dataset, encompassing (1) text-only generation: given the next image, generating the related text; (2) image-only generation: given the next text, generating the related image, and (3) multimodal generation: generating text-image pair by given context. The multimodal learning stage loss is given by $L_{\text{MLS}} = \lambda_1 * L_{\text{text}} + \lambda_2 * L_{\text{LDM}}$. More implementation details can be found in appendix B.

## 3 EXPERIMENTS

To assess the efficacy of our model, we conducted a series of evaluations across multiple benchmarks. These experiments aim to address several key questions: (1) Can our model generate plausible images and reasonable texts? (2) How does our model's performance stack up against other state-of-the-art models in both single-turn and multi-turn interleaved vision-and-language generation tasks? (3) What impact does the design of each module have on overall performance? In the subsequent subsections, we will delve into the datasets and experimental settings used for these evaluations, followed by a comprehensive analysis of our model's performance. More details about datasets and data format can be found in appendix C.

### 3.1 EXPERIMENTAL SETTINGS

**Baselines** For a comprehensive evaluation of our performance in multimodal generation, we conducted comparative analyses with several prominent baseline models: the Finetuned Unimodal Generation Model, GILL, and Divter.

- **Finetuned Unimodal Generation Model**: To facilitate fair comparisons in both image and text generation, we fine-tuned two separate models, Stable Diffusion 2 and MiniGPT-

Table 1: Performance metrics for different models with various prompt types on VIST final step image generation. For 'No Context', only the current step's text is provided. The 'Text Context' uses all history texts, the 'Image Context' employs all preceding images, and 'Image-Text Context' provides a combination of both past images and texts.

| Model | No Context | | | Text Context | | | Image Context | | | Image-Text Context | | |
|---|---|---|---|---|---|---|---|---|---|---|---|---|
| | CLIP-I (↑) | IS (↑) | FID (↓) | CLIP-I (↑) | IS (↑) | FID (↓) | CLIP-I (↑) | IS (↑) | FID (↓) | CLIP-I (↑) | IS (↑) | FID (↓) |
| Zero-shot SD 2 | 0.57 | **23.62** | 61.26 | 0.59 | 23.24 | 62.60 | - | - | - | - | - | - |
| Fine-tuned SD 2 | 0.59 | 23.28 | **58.29** | 0.61 | 23.47 | **57.45** | - | - | - | - | - | - |
| MiniGPT-5 (Prefix) | 0.60 | 23.19 | 61.25 | 0.63 | **25.06** | 61.81 | 0.68 | 24.27 | 59.92 | 0.70 | **25.10** | 60.46 |
| MiniGPT-5 (LoRA) | **0.61** | 22.30 | 61.44 | **0.64** | 23.86 | 61.34 | **0.69** | **25.03** | **59.09** | 0.70 | 24.38 | **59.48** |
| MiniGPT-5 (w/o UAS) | 0.55 | 16.32 | 73.02 | 0.57 | 16.31 | 73.97 | 0.58 | 16.70 | 75.88 | 0.58 | 16.99 | 76.51 |

4, utilizing the VIST dataset. Within the Stable Diffusion 2 model, the Unet parameters were unfrozen, and for MiniGPT-4's LLM part, LoRA parameters were incorporated.

- **GILL** (Koh et al., 2023)[1]: GILL is a recent innovation that allows the LLM to generate vokens using a pre-trained text-to-image generation model for single-image generation. Unlike our method, which employs conditional generation loss guidance, GILL minimizes the Mean Squared Error (MSE) loss between the text-to-image text encoding feature and voken features, similar to $L_{cap}$ in our approach. Since their method requests image descriptions for training, we compare with it just on the unimodal alignment stage.

- **Divter** (Sun et al., 2021): Divter is a state-of-the-art conversational agent developed for multimodal dialogue contexts. It introduces a customized transformer structure for generating multimodal responses. Divter's methodology includes pretraining on a vast corpus of text-only dialogues and text-image pairs, followed by finetuning on a select set of multimodal response data. The MMDialog dataset regards Divter's method as the baseline.

**Metrics** To comprehensively assess model performance across image, text, and multimodal dimensions, we employ a diverse set of metrics. For evaluating the quality and diversity of generated images, we utilize the Inception Score (IS) (Salimans et al., 2016) and Fréchet Inception Distance (FID) (Heusel et al., 2017). Textual performance is gauged through metrics such as BLEU (Papineni et al., 2002), Rouge-L (Lin, 2004), METEOR (Banerjee & Lavie, 2005), and Sentence-BERT (Reimers & Gurevych, 2019) scores.

On the multimodal front, we leverage CLIP-based metrics (Rombach et al., 2021) to assess the congruence between generated content and ground truth. CLIP-I evaluates the similarity between generated and ground-truth images, while CLIP-T focuses on the congruence between generated images and ground-truth text. To address potential misalignments in the multimodal generation, such as when the ground truth is text-only, but the output is multimodal, we utilize MM-Relevance (Feng et al., 2022). This metric calculates the F1 score based on CLIP similarities, providing a nuanced evaluation of multimodal coherence.

Recognizing that the generated multimodal output might be meaningful yet differ from the ground truth, we also incorporate human evaluation to assess the model's performance. We examine the model's effectiveness from three perspectives: (1) Language Continuity - assessing if the produced text aligns seamlessly with the provided context, (2) Image Quality - evaluating the clarity and relevance of the generated image, and (3) Multimodal Coherence - determining if the combined text-image output is consistent with the initial context.

## 3.2 EXPERIMENTAL RESULTS

In this section, we will quantitatively analyze our model performance on different benchmarks for different training stages. The qualitative examples can be found in Fig. 3.

---

[1]To ensure fair comparisons, given the variations in the valid data within the CC3M dataset and the original use of Stable Diffusion 1.5 in GILL, we made adjustments. Specifically, we switched their text-to-image generation model to Stable Diffusion 2 and retrained it on our specific CC3M data, following the guidelines in their official implementation. (https://github.com/kohjingyu/gill)

Table 2: VIST all steps image generation: CLIP Image-Image and FID Performance Metrics. In zero-shot SD2, for 'No Context', only the current step's text is provided. The 'Text Context' uses all historical texts. FID scores evaluate the similarities between generated images and ground truth images within each story sequence.

| Model | CLIP-I ($\uparrow$) | FID ($\downarrow$) |
|---|---|---|
| Zero-shot SD 2 (no/text context) | 0.58/0.59 | 414.34/393.49 |
| Fine-tuned SD 2 (no/text context) | 0.60/0.61 | 397.05/390.25 |
| MiniGPT-5 (Prefix) | 0.65 | 381.55 |
| MiniGPT-5 (LoRA) | **0.66** | **366.62** |
| MiniGPT-5 (w/o UAS) | 0.57 | 420.79 |

Table 3: VIST all steps narration generation: Sbert, Rouge-L, and Meteor Performance Metrics. We added LoRA fine-tuning for both MiniGPT-4 and MiniGPT-5. The results show that adding generative vokens does not hurt the performance on the multimodal comprehension tasks.

| Model | Sbert ($\uparrow$) | Rouge-L ($\uparrow$) | Meteor ($\uparrow$) |
|---|---|---|---|
| Fine-tuned MiniGPT-4 | 0.6273 | **0.3401** | **0.3296** |
| MiniGPT-5 | **0.6315** | 0.3373 | 0.3263 |

### 3.2.1 MULTIMODAL LEARNING STAGE

In this subsection, we present the performance of different models on the VIST (Huang et al., 2016) and MMDialg (Feng et al., 2022) datasets. Our evaluations span both vision (image-related metrics) and language (textual metrics) domains to showcase the versatility and robustness of the proposed models.

**VIST Final-Step Evaluation** Our first set of experiments involves a single-step evaluation where, given the last step's prompt, the model aims to generate the corresponding image. Table 1 summarizes the results for this setting. The MiniGPT-5 in all three settings can outperform the fine-tuned SD 2, showing the benefits of the MiniGPT-5 pipeline. Notably, the MiniGPT-5 (LoRA) model consistently surpasses other variants in terms of the CLIP Score across multiple prompt types, especially when both image and text prompts are combined. On the other hand, the FID scores highlight the MiniGPT-5 (prefix) model's competitiveness, indicating a possible trade-off between image embedding quality (reflected by the CLIP Score) and the diversity and realism of the images (captured by the FID score). When compared to the model (MiniGPT-5 w/o UAS) that undergoes direct training on the VIST without incorporating the unimodal alignment stage, it is evident that while the model retains the capability to generate meaningful images, there is a notable decline in image quality and coherence. This observation underscores the significance of our two-stage training strategies.

**VIST Multi-Step Evaluation** In a detailed and comprehensive evaluation, we systematically provided models with prior history context and subsequently assessed the generated images and narrations at each following step. Tables 2 and 3 outline the results of these experiments, encapsulating the performance in both image and language metrics, respectively. The findings demonstrate that MiniGPT-5 is capable of generating coherent, high-quality images utilizing long-horizontal multimodal input prompts across all data, without compromising the original model's ability for multimodal comprehension. This accentuates the efficacy of our model in diverse settings.

**VIST Human Evaluation** To assess the quality of multimodal generation, we tested both our model and the baseline on the VIST validation set. For each task, given a preceding multimodal sequence, models are tasked with producing the subsequent scenario. To ensure a fair comparison, we employed the fine-tuned MiniGPT-4, which is exclusively trained to generate narrations without any vokens. Subsequently, these narrations are incorporated directly into the Stable Diffusion 2 via the text-to-image pipeline. We selected a random sample of 5,000 sequences, with each requiring evaluation by two workers. These evaluators are tasked with determining the superior multimodal output based on three criteria: Language Continuity, Image Quality, and Multimodal Coherence. This assessment was facilitated using Amazon Mechanical Turk (Crowston, 2012), with a representative example (Fig. 4) provided in the appendix. As depicted in Table 4, our model, MiniGPT-5,

Table 4: VIST Human Evaluation on 5000 samples for multimodal generation from Language Continuity, Image Quality, and Multimodal Coherence aspects. The results indicate, in more than 70% cases, the MiniGPT-5 is better or on par with the two-stage baseline.

| Model | MiniGPT-5 | Fine-tuned MiniGPT-4 + SD 2 | Tie |
|---|---|---|---|
| Language Continuity (%) | **57.18** | 28.51 | 14.31 |
| Image Quality (%) | **52.06** | 35.98 | 11.96 |
| Multimodal Coherence (%) | **57.62** | 23.24 | 19.14 |

Table 5: Multimodal generation results on MMDialog test set. In order to compare with their baseline, we use the same metrics reported in Table 3 of MMDialog paper (Feng et al., 2022).

| Model | IS (↑) | BLEU-1 (↑) | BLEU-2 (↑) | Rouge-L (↑) | MM-Relevance (↑) |
|---|---|---|---|---|---|
| Divter | **20.53** | 0.0944 | 0.0745 | 0.1119 | 0.62 |
| MiniGPT-5 | 19.63 | **0.2221** | **0.1546** | **0.1119** | **0.67** |

was found to generate more fitting text narrations in 57.18% of instances, deliver superior image quality in 52.06% of cases, and produce more coherent multimodal outputs in 57.62% of the scenarios. This data distinctly showcases its enhanced multimodal generation capabilities when compared to the two-stage baseline that employs narrations for text-to-image prompts without the inclusion of vokens.

**MMDialog Multi-Turn Evaluation**    We conducted an evaluation of our method on the MMDialog dataset to determine the effectiveness of generating precise and appropriate multimodal information in multi-turn conversational scenarios. The model is required to generate either unimodal or multimodal responses based on the previous turns during the conversations in this dataset. Our results, as presented in Table 5, demonstrate that MiniGPT-5 outperforms the baseline model Divter in terms of generating more accurate textual responses. While the image qualities of the generated responses are similar, MiniGPT-5 excels in MM-Relevance compared to the baseline model. This indicates that our model can better learn how to appropriately position image generation and produce highly coherent multimodal responses.

### 3.2.2 Unimodal Alignment Stage

Instead of evaluating on the datasets with multi-turn multimodal data, we also evaluate models in the single-image dataset CC3M (Sharma et al., 2018), as displayed in Table 6. The results indicate that although our model can have better generation on multi-turn scenarios, the Stable Diffusion 2 model achieves the best outcomes across all metrics for single-image generation. Since our model attempts to align with the pretrained text encoder of Stable Diffusion 2 in this stage, there is a slight gap in performance due to the limitation of data amount. Compared with the observations on the VIST dataset, we can conclude that MiniGPT-5 can correctly extract features from long-horizontal multimodal information instead of single text input. This indicates the future directions on how to align LLMs with generative models efficiently. On the other hand, our model outperforms another state-of-the-art multimodal generation model, GILL, on all metrics. Our model generates more coherent and high-quality images that closely resemble those produced by the pretrained stable diffusion model. To further evaluate the effectiveness of our design, we conducted several ablation studies, and more ablation studies about voken number and CFG scales can be found in appendix D.

**Evaluation of Different Loss Guidance:**    We introduced an auxiliary loss, denoted as $L_{cap}$ for CC3M training. To assess the impact of this loss and determine if the single caption loss alone can generate high-quality images like GILL, we trained our model without the caption loss $L_{cap}$ and the conditional latent diffusion loss $L_{LDM}$ separately. The results, as shown in Table 6, indicate that the caption loss significantly aids in generating better images, and the conditional latent diffusion loss further enhances performance in terms of coherence and image quality.

**Evaluation of Classifier-Free Guidance (CFG):**    To assess the effectiveness of the CFG strategy, we trained our model without CFG dropoff. During inference, the model utilized the original

Table 6: Model performance on CC3M validation set for single-image generation. Due to the limitations of data amount, we find there is still a gap for voken alignment with Stable Diffusion 2. However, our model outperforms another state-of-the-art model, GILL, in all metrics.

| Model | CLIP-I (↑) | CLIP-T (↑) | IS (↑) | FID (↓) |
|---|---|---|---|---|
| Zero-shot SD 2 | **0.64** | **0.25** | **31.74** | **26.39** |
| GILL | 0.57 | 0.20 | 22.76 | 37.97 |
| MiniGPT-5 | 0.61 | 0.22 | 28.09 | 31.47 |
| MiniGPT-5 (w/o CFG) | 0.60 | 0.22 | 23.41 | 33.73 |
| MiniGPT-5 (w/o $L_{cap}$) | 0.54 | 0.16 | 21.27 | 40.24 |
| MiniGPT-5 (w/o $L_{LDM}$) | 0.58 | 0.20 | 24.79 | 34.65 |

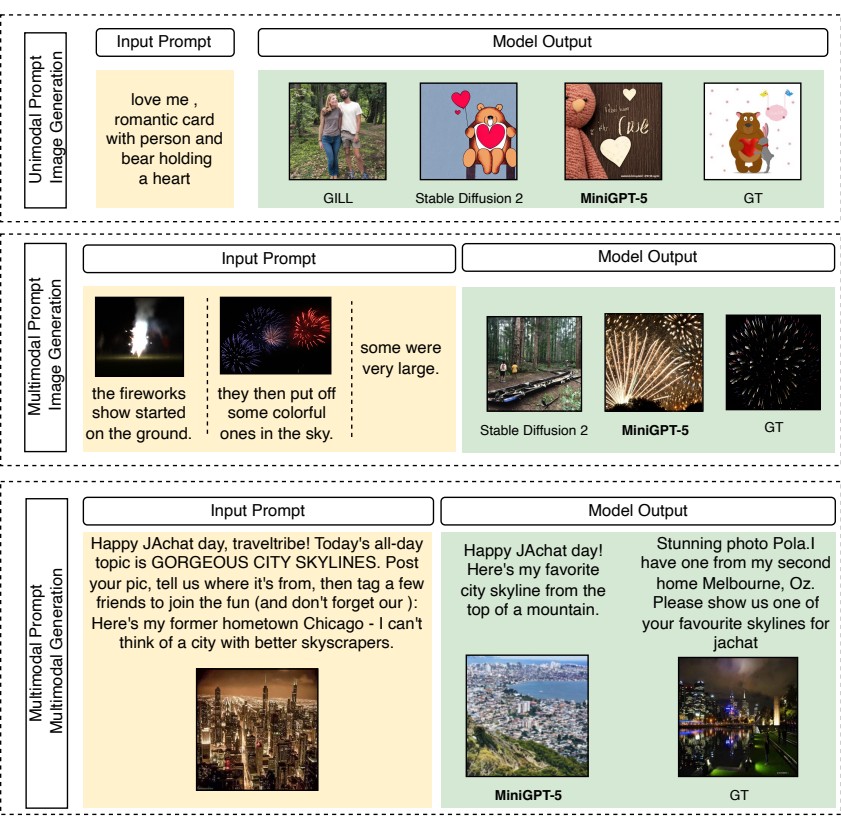

Figure 3: Qualitative examples from MiniGPT-5 and baselines. From the comparisons, we can find the MiniGPT-5 and SD 2 have similar results on single-image generation. When we evaluate with multi-step multimodal prompts, MiniGPT-5 can produce more coherent and high-quality images. More qualitative examples can be found in the appendix E.

CFG denoising process, which utilized the empty caption feature from SD 2's text encoder as negative prompt features. The results in Table 6 demonstrate that all metrics are worse without CFG, indicating that the CFG training strategy improves image generation quality.

## 4 CONCLUSION

In this paper, we introduce a novel model structure, MiniGPT-5, designed to augment the capabilities of LLMs for multimodal generation by aligning the LLM with a pre-trained text-to-image generation model. Our approach demonstrates substantial improvements, as evidenced by comprehensive experiments. Through this work, we aspire to set a new benchmark in multimodal generative models, opening doors to applications previously deemed challenging due to the disjointed nature of existing image and text synthesis paradigms.

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
