## A  RELATED WORK

**Text-to-Image Generation**   To transform textual descriptions into their corresponding visual representations, text-to-image models (Reed et al., 2016; Dhariwal & Nichol, 2021; Saharia et al., 2022; Rombach et al., 2021; 2022) employ complex architectures and sophisticated algorithms, bridging the gap between textual information and visual content. These models are adept at interpreting the semantics of input text and translating them into coherent and pertinent images. A notable recent contribution in this field is Stable Diffusion 2 (Rombach et al., 2021), which employs a diffusion process to generate conditional image features and subsequently reconstructs images from these features. Our research aims to leverage this pre-trained model, enhancing its capabilities to accommodate both multimodal input and output.

**Multimodal Large Language Models**   As Large Language Models (LLMs) become increasingly impactful and accessible, a growing body of research has emerged to extend these pretrained LLMs into the realm of multimodal comprehension tasks (Zhu et al., 2023; Li et al., 2023b; Dai et al., 2023; OpenAI, 2023; Li et al., 2023a; Alayrac et al., 2022). For example, to reproduce the impressive multimodal comprehension ability in GPT-4 (OpenAI, 2023), MiniGPT-4 (Zhu et al., 2023) proposes a projection layer to align pretrained vision component of BLIP- (Li et al., 2023b) with an advanced open-source large language model, Vicuna (Chiang et al., 2023). In our work, we utilize the MiniGPT-4 as the base model and extend the model's capabilities to multimodal generation.

**Multimodal Generation with Large Language Models**   To augment the LLM's capabilities in seamlessly integrating vision and language generation, recent studies have introduced a variety of innovative methods (Ge et al., 2023; Sun et al., 2021; Koh et al., 2023; Sun et al., 2023b; Yu et al., 2023). For instance, CM3Leon (Yu et al., 2023) presents a retrieval-augmented, decoder-only architecture designed for both text-to-image and image-to-text applications. Similarly, Emu (Sun et al., 2023b) employs the pretrained EVA-CLIP (Sun et al., 2023a) model to convert images into one-dimensional features and fine-tunes the LLAMA (Touvron et al., 2023) model to generate cohesive text and image features through autoregressive techniques. On the other hand, both GILL (Koh et al., 2023) and SEED (Ge et al., 2023) explore the concept of mapping vokens into the text feature space of a pretrained Stable Diffusion model; GILL employs an encoder-decoder framework, while SEED utilizes a trainable Q-Former structure. In contrast to these approaches, our model takes a more direct route by aligning voken features with visual information. Additionally, we introduce several training strategies aimed at enhancing both image quality and contextual coherence.

## B  IMPLEMENTATION DETAILS

In the unimodal alignment stage, we introduce additional voken embeddings at both the input and output layers of the Vicuna model, while keeping the embeddings of other tokens fixed. These new embeddings—denoted as $\theta_{\text{voken\_input}}$ and $\theta_{\text{voken\_output}}$—along with the feature mapper module $(\theta_{\text{MLP}}, \theta_{\text{enc\_dec}}, q)$ are jointly trained on the CC3M dataset, which consists of single text-image pairs. Training is conducted using the AdamW optimizer over two epochs, with a batch size of 48, amounting to over 110,000 steps, and a learning rate of $2 \times 10^{-4}$.

In the subsequent multimodal learning stage, we incorporate LoRA modules—denoted as $\theta_{\text{LoRA}}$—into Vicuna for the generation of both tokens and vokens. We keep the MLP model $\theta_{\text{MLP}}$ and decoder query $q$ fixed. The model is then fine-tuned on interleaved vision-and-language datasets, like VIST and MMDialog. The trainable parameters for this stage are $\theta = \{\theta_{\text{voken\_input}}, \theta_{\text{voken\_output}}, \theta_{\text{LoRA}}, \theta_{\text{enc\_dec}}\}$. Training is carried out using the AdamW optimizer over four epochs, with a batch size of 32 and a learning rate of $2 \times 10^{-5}$.

## C  EXPERIMENTAL SETTINGS

### C.1  DATASETS

**CC3M (Sharma et al., 2018):**   Conceptual Captions dataset represents a remarkable collection of high-quality image captions, amassing approximately 3.3 million pairs of text and images from

the internet. The CC3M dataset's diverse content, quality assurance, and support for multimodal learning make it a valuable asset for researchers and AI enthusiasts alike. Each data sample within this dataset consists of an image accompanied by a corresponding text description, reflecting the richness of human language and visual perception. However, after accounting for license restrictions and eliminating invalid image links, the dataset now comprises approximately 2.2 million data pairs suitable for training purposes and 10 thousand data pairs designated for validation.

**VIST (Huang et al., 2016):** Visual Storytelling dataset is an innovative compilation of visual narratives. The VIST dataset's engaging content, narrative structure, and emphasis on sequential understanding position it as an essential resource for researchers focusing on sequential image understanding. Each sequence within this dataset consists of five images accompanied by corresponding textual narratives, showcasing the intricate interplay between visual imagery and storytelling. Designed to foster creativity and challenge conventional image-captioning models, the dataset provides a platform for training and validating algorithms capable of generating coherent and contextually relevant stories. After eliminating the invalid image links, we got over 65 thousand unique photos organized into more than 34 thousand storytelling sequences for training and 4 thousand sequences with 8 thousand images for validation.

**MMDialog (Feng et al., 2022):** Multi-Modal Dialogue dataset stands as the largest collection of multi-modal conversation dialogues. The MMDialog dataset's extensive scale, real human-human chat content, and emphasis on multi-modal open-domain conversations position it as an unparalleled asset for researchers and practitioners in artificial intelligence. Each dialogue within this dataset typically includes 2.59 images, integrated anywhere within the conversation, showcasing the complex interplay between text and visual elements. Designed to mirror real-world conversational dynamics, the dataset serves as a robust platform for developing, training, and validating algorithms capable of understanding and generating coherent dialogues that seamlessly blend both textual and visual information.

## C.2 DATA FORMAT

**Unimodal Alignment Stage** In the unimodal alignment stage, our objective is to synchronize the generative voken with the text-to-image model's conditional feature, focusing on single-turn text-image pairs. To achieve this, we utilize data from the CC3M dataset, constructing training samples by appending vokens as image placeholders after the captions, such as "a big black dog [IMG1] ... [IMGn]." The Language Model (LLM) is then tasked with only generating these placeholders for text creation, and the corresponding output hidden features are further employed to compute the conditional generation loss with the ground truth image.

**Multimodal Learning Stage** In this stage, we utilize the VIST and MMDialog datasets, both of which contain multi-turn multimodal data. During training, we integrate placeholders for input images, such as '<ImageHere></Img>', into the input text prompts when applicable. These prompts also encompass various instructions corresponding to different task types, with outputs manifesting as pure-text, pure-voken, or text-voken combinations. Below, we present example templates in the VIST dataset to illustrate the different task types:

- **Text Generation:** Input: "<History Context> What happens in the next scene image: <ImageHere></Img>"; Output: "<Text Description>"

- **Image Generation:** Input: "<History Context> Generate an image with the scene description: [Text Description]"; Output: "[IMG1]...[IMGn]"

- **Text-Image Generation:** Input: "<History Context> What should happen then?"; Output: "<Text Description> [IMG1]...[IMGn]"

By structuring the input and output in this manner, we create a flexible framework that accommodates various multimodal tasks, enhancing the model's ability to interpret and generate both textual and visual content. In the VIST dataset, the history context includes all previous story steps with both texts and images. In the MMDialog dataset, due to the limitation of computational resources, we only use up to one previous turn as the history context, and all data are formatted into the dialog.

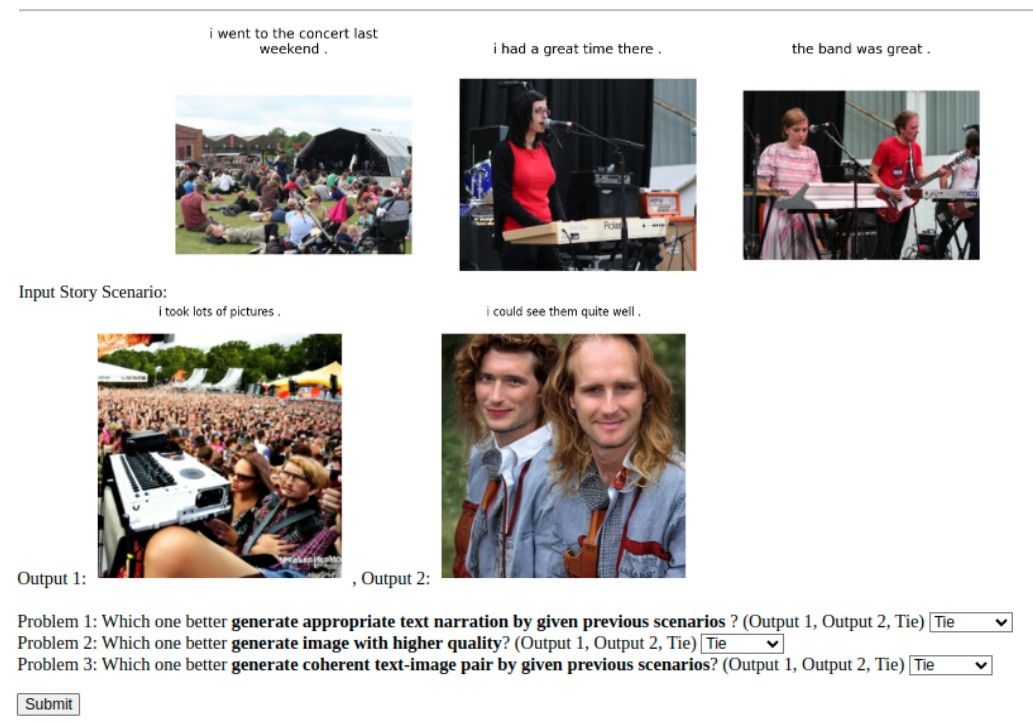

Figure 4: Screenshot for human evaluation interface on Amazon Mechanical Turk. Output 1 is generated by MiniGPT-5, while output 2 is generated by fine-tuned MiniGPT-4 (without vokens) and stable diffusion 2.

## D  MORE EXPERIMENTS

### D.1  EVALUATION OF GUIDANCE SCALE:

Since our model incorporates CFG, it is crucial to evaluate how different guidance scales affect image generation. Therefore, we plotted several line charts in Fig 5 to depict the changes in metrics with varying guidance scales. The figures reveal that both the stable diffusion model and our model generate better images as the guidance scale increases. However, when the scale exceeds 10, the image semantic coherence stabilizes while the image quality continues to decline. This suggests that the guidance scale should be set within a reasonable range for optimal image generation.

### D.2  EVALUATION OF VOKEN NUMBER:

The voken features in our model are directly utilized as conditions in the text-to-image model, leading to the expectation that an increase in the number of vokens would enhance the model's representative capabilities. To validate this hypothesis, we conducted an experiment by training the model with varying numbers of vokens, ranging from 1 to 16. As illustrated in Fig 6, the model's performance consistently improves with the addition of more vokens. This improvement is particularly noticeable when the number of vokens is increased from 1 to 4, highlighting the significant role that vokens play in enhancing the model's effectiveness.

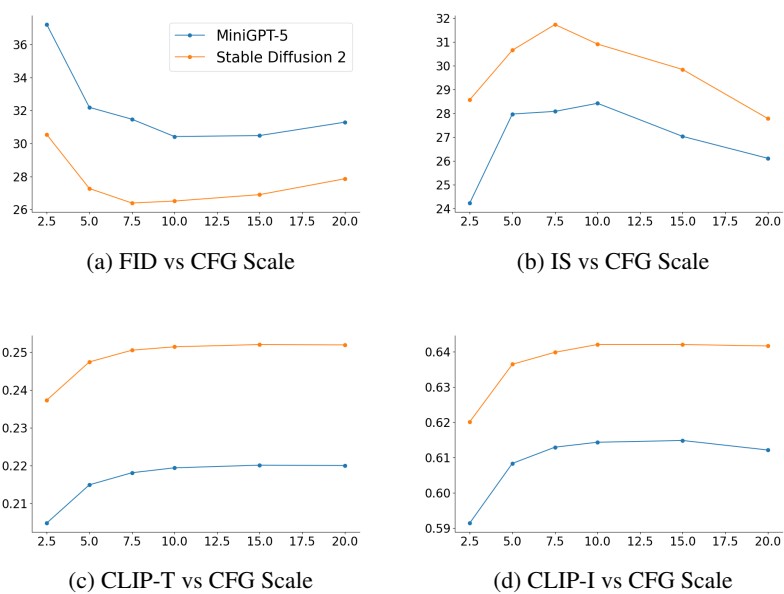

(a) FID vs CFG Scale

(b) IS vs CFG Scale

(c) CLIP-T vs CFG Scale

(d) CLIP-I vs CFG Scale

Figure 5: Line charts for various metrics vs Classifier-free Guidance (CFG) scale. The results suggest that our CFG strategy can exhibit comparable effectiveness to the CFG strategy employed in SD2, with the appropriate CFG scale significantly enhancing both image quality and coherence.

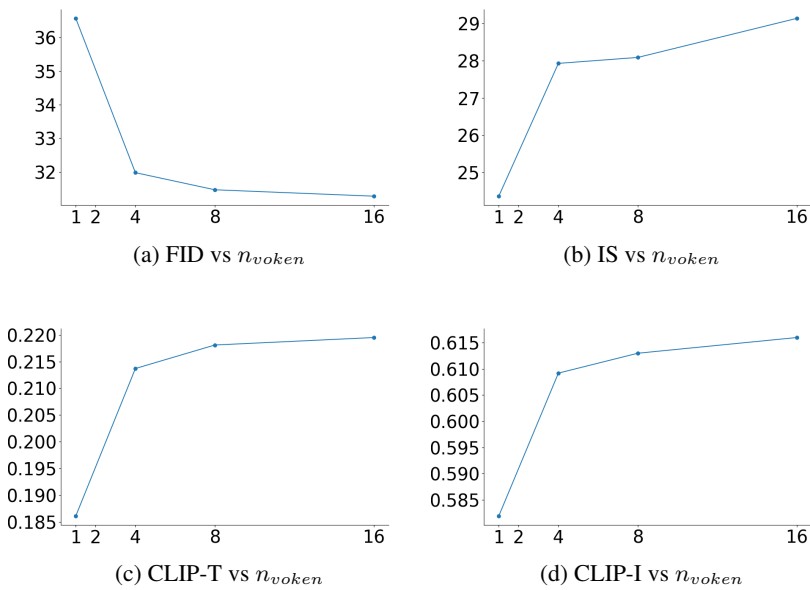

(a) FID vs $n_{voken}$

(b) IS vs $n_{voken}$

(c) CLIP-T vs $n_{voken}$

(d) CLIP-I vs $n_{voken}$

Figure 6: Line charts for various metrics vs the number of vokens. As the number of vokens increases, the image quality and CLIP scores improve. In this work, our default voken number is 8.

# E MORE QUALITATIVE EXAMPLES

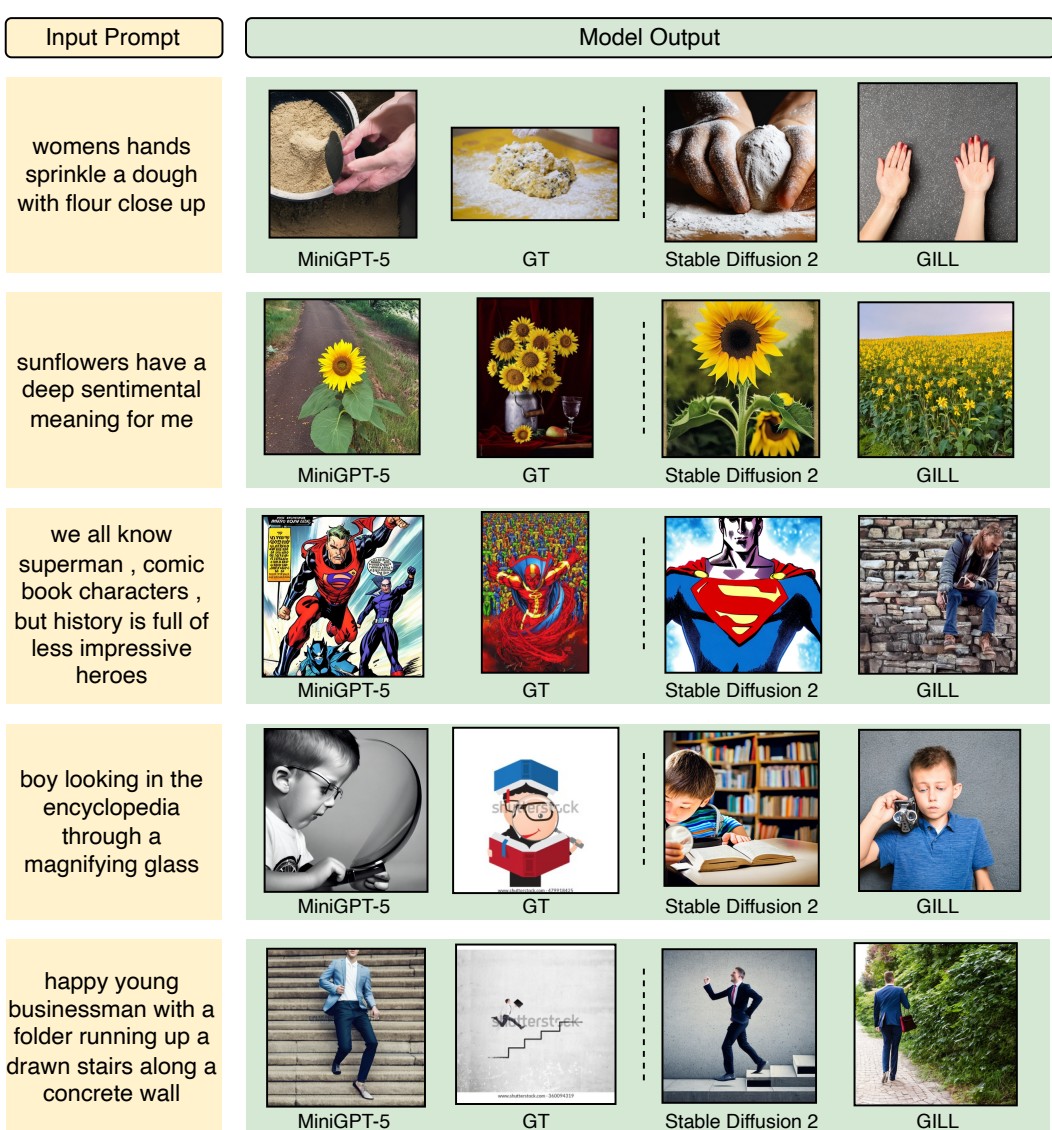

Figure 7: More qualitative examples from MiniGPT-5 and baselines on CC3M validation set.

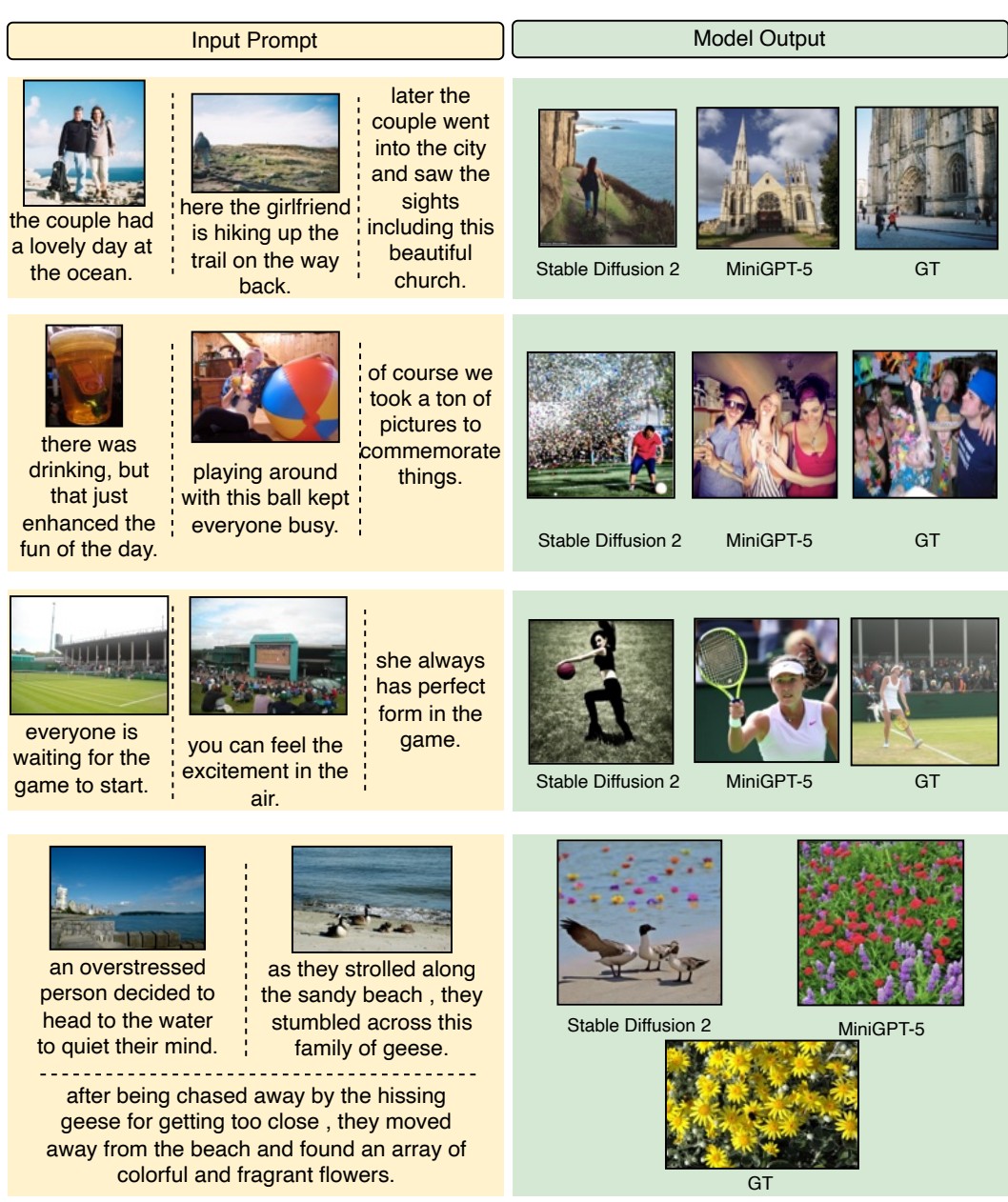

Figure 8: More qualitative examples from MiniGPT-5 and baselines on VIST validation set.

| Input Prompt | Model Output |
|---|---|

Happy Traveltuesday Where would you rather be today? I'd love to be back enjoying the beautiful St Monans night skies ScotSpirit

I'd love to be back in St Monans night skies ScotSpirit!

**MiniGPT-5**

One Day I'll Visit Scotland So on this TravelTuesday - Highlands or Sea - Somewhere in Scotland is Where I'd Like to Be

**GT**

Also, Eli Whitmire a very deserving offensive MVP award.

Eli Whitmire is the best player in college football.

**MiniGPT-5**

my fav pic of eli Whitmire. roasted

**GT**

One activity that many can still do right now is hiking. :raising_hands: If you've got a second, I'd love to hear about your favourite hike you've ever done! :hiking_boot: :mountain: Feel free to drop a link to an article as well. Thought it'd be fun to create a little hiking bucket list for the future!

I've been to the top of Mount Washington in New Hampshire. It was a tough climb but worth it!

**MiniGPT-5**

One of the best hikes in Canada (and not at all hard) is the beautiful Skerwink Trail in Port Rexton, Newfoundland/Labrador. When you finish you can take a short walk to the Port Rexton Brewing Co. brewpub! Canada

**GT**

Figure 9: More qualitative examples from MiniGPT-5 on MMDialog test set.