# OpenReview forum: "MiniGPT-5: Interleaved Vision-and-Language Generation via Generative Vokens"
_ICLR.cc/2024/Conference — ICLR 2024 Conference Withdrawn Submission_

### Official Review · Reviewer_sgCo · 2023-11-01

**Soundness:** 1 poor
**Presentation:** 2 fair
**Contribution:** 2 fair
**Rating:** 3
**Confidence:** 5

**Summary:**

The paper proposed to leverage LLMs to process and generate multimodal contents. Specifically, the paper leverages the pretrained MiniGPT-4, and extend it to generate interleaved image/text contents, by finetuning its output to contain both text tokens and visual tokens, where the visual tokens are fed to Stable Diffusion models in place of the original text-condition features to generate the images. The paper finetunes MiniGPT-4 on VIST and MMDialog dataset and demonstrates better results than the baselines.

**Strengths:**

- Enhancing multimodal models to generate multimodal content is an important enhancement in capabilities with many potential applications.
- The proposed model achieves a better performance than the baselines on two evaluated datasets.

**Weaknesses:**

- It is unclear why GILL cannot be compared on VIST and MMDialog.
  * GILL trains on CC3M image-text pairs dataset, and the images and captions are also used in this paper.
  * (please correct me if I am wrong) GILL does not finetune on VIST / MMDialog while this paper does.

- Lack of the synthetic caption baseline.
  * Despite the paper argues the scarcity of the descriptive captions in the existing datasets and it is hard to automatically generate high-quality captions, there is no quantitative numbers justifying that the synthetic caption has poor quality.

- The baseline experiments are not properly explained, and there can be as simple but stronger baselines for comparison
  * Comparison with MiniGPT-4: the authors do not clearly explain the inputs and outputs used to finetune MiniGPT-4. A proper baseline would be to use MiniGPT-4 to caption the training images and train it to output both a text (story) and a description (prompt for SD).

- The design choices are not well-ablated/well-justified.

  * The proposed voken dropping for CFG is not ablated. The simplest alternative is to just use the default empty feature of the SD in inference. An alternative is to learn the empty features, without messing the weights in the Feature Mapper.

  * Why the number of vokens $n=8$? In Supp. Fig. 6, $n=16$ is consistently better than $n=8$ (significantly better in $IS$). Also, given the trend (especially given the large leap for IS from 8$\rightarrow$16), why not ablating $n>16$? Furthermore, there are 77 tokens for the text encoder in Stable Diffusion -- in Table 6, MiniGPT-5 performs worse than Stable Diffusion, is this because $n$ is too small?

- The name of the proposed approach is confusing and needs justification.
The proposed method is named MiniGPT-5. However, the naming is confusing and needs better justification -- there is no GPT-5 from OpenAI available yet. The naming itself can make people confused on whether OpenAI has released GPT-5 or not. Also, it is unclear why the authors name the approach this way.

**Questions:**

### Questions

> Two-stage Training Strategy: After the unimodal alignment stage, the model is capable of generating images for single text descriptions but struggles with interleaved vision-and-language generation, which includes multiple text-image pairs and requires complicated reasoning for both text and image generation.

Is this because the base MiniGPT-4 is not trained to understand the interleaved image-text conversations? Is the second-stage training still needed if the base VLM is capable of understanding / processing interleaved image/text pairs?

> The inclusion of classifier-free guidance during the training phase further re- fines generation quality.

Can we really say that we include CFG during the training? Token dropping is not the same as CFG.

> We also established unprecedented benchmarks on prominent datasets, including VIST and MMDialog.

It would be better justified on *unprecedented* benchmarks, if the authors can provide more substantitated explanations / justifications.

### Minor Questions

> We employ the loss of the latent diffusion model (LDM) for guidance.

The term "guidance" in the sentence creates ambiguity, especially when juxtaposed with terms like "classifier guidance" and "classifier-free guidance."

---

### Official Review · Reviewer_vkr4 · 2023-11-01

**Soundness:** 2 fair
**Presentation:** 3 good
**Contribution:** 2 fair
**Rating:** 5
**Confidence:** 4

**Summary:**

The paper introduces an interleaved vision-and-language generation model named MiniGPT-5. Architecturally, the model employs PEFT on MiniGPT-4 to generate text and vokens, and use a feature mapper consisting of an encoder-decoder to map vokens to the conditions of SD, which then generates the final image. For training, a Two-stage Training Strategy is used, starting with the Unimodal Alignment Stage (UAS) followed by the Multimodal Learning Stage (MLS). The model was evaluated on the MMDialog and VIST datasets.

**Strengths:**

1. Generating text and images under Language and Image Context is an intriguing task with potential real-world applications.
2. The proposed method leverages pre-trained text and image generation models, requiring only paramter-efficient fine-tuning the language model and the training of the Feature Mapper, resulting in reduced training overhead.

**Weaknesses:**

1. Based on the experimental results, the proposed approach did not exhibit significant improvements on the VIST dataset compared to standalone SD. On the MMDialog dataset, its Inception Score (IS) was lower than that of Divter. The quality of showcased generated image (such as human) is also not satisfying
2. Several existing methods can achieve interleaved vision-and-language generation, such as Visual ChatGPT. A straightforward method might involve a language model determining if image generation is needed, then producing a descriptive segment about the image to input into Stable Diffusion. The paper does not compare against such methods nor articulates advantages over them.

**Questions:**

1. A potential advantage of this method may lie in its better understanding of image context, linking prior images to generate new ones. An example is editing images based on prior images and language instructions. Can the authors provide examples in the realm of image editing?
2. In the results for the VIST dataset (Table1), (1) With Text Context, MiniGPT-5 underperforms compared to Finetuned SD; (2) MiniGPT-5's FID in *Image-Text Context* is even worse than in *Image-Context*. Why?
3. Can the authors provide additional examples, particularly multi-turn image-text dialogues?

**Details Of Ethics Concerns:**

1. The experimental results are not compelling.
2. The work lacks comparison with sufficient kinds of interleaved vision-and-language generation methods. It does not display evident advantages over methods like Visual ChatGPT.

---

### Official Review · Reviewer_zEXf · 2023-11-01

**Soundness:** 3 good
**Presentation:** 2 fair
**Contribution:** 3 good
**Rating:** 5
**Confidence:** 3

**Summary:**

This paper introduces miniGPT-5, a model capable of generating multimodal outputs, including both text and images. The authors integrate a vision encoder with Large Language Models (LLMs) to produce vokens, facilitating multimodal generation.

**Strengths:**

The concept of the paper is clear and straightforward, presenting a meaningful advancement in the integration of multimodal inputs and outputs within LLMs.

The authors have conducted comprehensive experiments across various cases, such as text-to-image, (text, image) to image, and (text, image) to (text, image), demonstrating the versatility of miniGPT-5.

**Weaknesses:**

The paper lacks clarity on how the rationality of human evaluation in Table 4 is assessed, raising questions about the validity and reliability of these subjective measures.

In terms of image generation performance, miniGPT-5 falls short of surpassing the results achieved by the SD2 model on the CC3M dataset.

The use of the title "miniGPT-5" could potentially lead to confusion within the community, as the structure and capabilities of GPT-5 remain undefined at this time.

**Questions:**

See Weaknesses

---

### Official Review · Reviewer_yx5e · 2023-11-03

**Soundness:** 3 good
**Presentation:** 2 fair
**Contribution:** 2 fair
**Rating:** 5
**Confidence:** 4

**Summary:**

This paper connects the pretrained MinIGPT-4 and text-to-image Stable Diffusion model via a feature mapper module for multimodal generation. The feature mapper module includes a two-layer MLP and an encoder-decoder transformer to transform the voken features as a conditional signal for the following SD-based image generation.

The authors also propose a two-stage training method, Unimodal Alignment Stage (UAS) and Multimodal Learning Stage (MLS). The former one utilizes image caption dataset to enhance the single text-based image generation capability. The latter one aims to improve the interleaved visual-textual generation.

The experiments on three datasets (VIST, MMDialog and CC3M) verify the proposed MiniGPT-5.

**Strengths:**

1) Multimodal generation is now a popular direction. This paper improves MiniGPT-4 so that the model can output text and images. Previous MiniGPT-4 can not generate visual content.

2) The proposed solution is simple and easy to understand. They combine MiniGPT-4 and SD model for generating text and image respectively. In order to improve the quality of multimodal generation, the authors propose an effective two-stage training method and some auxiliary losses.

3) The authors provide extensive analysis for their experimental results.

**Weaknesses:**

1) Unfair comparison.

a) [VIST Human Evaluation] The previous MiniGPT-4 itself does not have multimodal generation capability, and it is unacceptable to directly use its output as the input of SD model. Thus, the results on Table 4 cannot illustrate the effectiveness of the proposed method.

b) It is necessary to list and compare model parameters and training data with some baselines, such as SD 2, MiniGPT-4, Divter, GILL.


2) The experimental results are not satisfactory.

The proposed method performs worse than fine-tuned MiniGPT-4 on narration generation, as shown in Table 3.

3) The presentation is general. Fig 2 can be improved.

Other comments:
1) Page 2: Stable DIffusion -> Stable Diffusion

2) Caption of Fig 2: Stable Duffision -> Stable Diffusion

3) Fig 2: Learable -> Learnable

**Questions:**

1) How to understand the learnable queries in Figure 2? Why is it input directly into transformer's decoder.

2) Why is the proposed method a description-free learning process?

3) There are many multimodal generation methods. Why did you not conduct a performance comparison, but focused on SD 2 and MiniGPT-4. Moreover, when comparing with GILL, the original experimental settings were not followed.

[1] Grounding Language Models to Images for Multimodal Inputs and Outputs

[2] NExT-GPT: Any-to-Any Multimodal LLM